# Optogenetic activation of heterotrimeric G-proteins by LOV2GIVe, a rationally engineered modular protein

Mikel Garcia-Marcos*, Kshitij Parag-Sharma[†], Arthur Marivin, Marcin Maziarz, Alex Luebbers, Lien T Nguyen

Department of Biochemistry, Boston University School of Medicine, Boston, United States

*For correspondence:
mgm1@bu.edu

Present address: [†]Graduate Curriculum in Cell Biology and Physiology, Biological andBiomedical Sciences Program, University of North Carolina, Chapel Hill, United States

Competing interests: The authors declare that no competing interests exist.

**Abstract** Heterotrimeric G-proteins are signal transducers involved in mediating the action of many natural extracellular stimuli and many therapeutic agents. Non-invasive approaches to manipulate the activity of G-proteins with high precision are crucial to understand their regulation in space and time. Here, we developed LOV2GIVe, an engineered modular protein that allows the activation of heterotrimeric G-proteins with blue light. This optogenetic construct relies on a versatile design that differs from tools previously developed for similar purposes, that is metazoan opsins, which are light-activated G-protein-coupled receptors (GPCRs). Instead, LOV2GIVe consists of the fusion of a G-protein activating peptide derived from a non-GPCR regulator of G-proteins to a small plant protein domain, such that light uncages the G-protein activating module. Targeting LOV2GIVe to cell membranes allowed for light-dependent activation of Gi proteins in different experimental systems. In summary, LOV2GIVe expands the armamentarium and versatility of tools available to manipulate heterotrimeric G-protein activity.

## Introduction

Heterotrimeric G-proteins are critical transducers of signaling triggered by a large family of G-protein-coupled receptors (GPCRs). GPCRs are Guanine nucleotide Exchange Factors (GEFs) that activate G-proteins by promoting the exchange of GDP for GTP on the Gα-subunits (*Gilman, 1987*). This signaling axis regulates a myriad of (patho)physiological processes and also represents the target for >30% of drugs on the market (*Sriram and Insel, 2018*), which fuels the interest in developing tools to manipulate G-protein activity in cells with high precision.

Optogenetics, the use of genetically-encoded proteins to control biological processes with light (*Warden et al., 2014*), is well-suited for the non-invasive manipulation of signaling. Metazoan opsins are light-activated GPCRs that have been repurposed as optogenetic tools (*Airan et al., 2009*; *Bailes et al., 2012*; *Karunarathne et al., 2013*; *Oh et al., 2010*; *Siuda et al., 2015*), albeit with limitations. For example, opsins tend to desensitize after activation due to receptor internalization and/or exhaustion of their chromophore, retinal (*Airan et al., 2009*; *Bailes et al., 2012*; *Siuda et al., 2015*). Exogenous supplementation of retinal, which is not always feasible, is required in many experimental settings because this chomophore is not readily synthesized by most mammalian cell types or by many organisms. Also, opsins are relatively large, post-translationally modified transmembrane proteins, which makes them inherently difficult to modify for optimization and/or customization for specific applications.

Here we leveraged the light-sensitive LOV2 domain of *Avena sativa* (*Harper et al., 2003*) to develop an optogenetic activator of heterotrimeric G-proteins not based on opsins. LOV2 uses ubiquitously abundant FMN as the chromophore and does not desensitize. It is small (~17 KDa) and expresses easily as a soluble globular protein in different systems (*Lungu et al., 2012*;

*Strickland et al., 2012*; *Zayner et al., 2013*), making it experimentally tractable, easily customizable, and widely applicable. Our results provide the proof-of-principle for a versatile optogenetic activator of heterotrimeric G-proteins that does not rely on GPCR-like proteins.

## Results and discussion

### Design and optimization of a photoswitchable G-protein activator

We envisioned the design of a genetically-encoded photoswitchable activator of heterotrimeric G-proteins by fusing a short sequence (~25 amino acids) from the protein GIV (a.k.a. Girdin) called the *Gα-Binding-and-Activating* (GBA) motif to the C-terminus of LOV2 (*Figure 1A*, left). GBA motifs are evolutionarily conserved sequences found in various cytoplasmic, non-GPCR proteins that are sufficient to activate heterotrimeric G-protein signaling (*DiGiacomo et al., 2018*). We reasoned that the GBA motif would be 'uncaged' and bind G-proteins when the C-terminal Jα-helix of LOV2 becomes disordered upon illumination (*Harper et al., 2003*; *Figure 1A*, right). We named this opto-genetic construct LOV2GIV (pronounced '*love-to-give*'). To facilitate the biochemical characterization of LOV2GIV, we initially used two well-validated mutants that mimic either the dark (D) or the lit (L) conformation of LOV2 (*Harper et al., 2004*; *Harper et al., 2003*; *Zimmerman et al., 2016*; *Figure 1A*, right), with the intent of subsequently validating the G-protein regulatory activity of LOV2GIVe in cells using light.

Our first LOV2GIV prototype had a suboptimal dynamic range, as it bound the G-protein Gαi3 in the lit conformation only ~3 times more than in the dark conformation (*Figure 1B*). Based on a structural homology model of LOV2GIV (*Figure 1C*), we reasoned that it could be due to the relatively high accessibility of Gαi3-binding residues of the GIV GBA motif (L1682, F1685, L1686) within the dark conformation of LOV2GIV. In agreement with this idea, fusion of the GBA motif at positions of the Jα helix more proximal to the core of the LOV2 domain tended to lower G-protein binding, including one variant ('e', henceforth referred to as LOV2GIV<u>e</u>) in which binding to the dark conformation was almost undetectable (*Figure 1D*). Consistent with this, a structure homology model of LOV2GIVe showed that amino acids required for G-protein binding are less accessible than in the LOV2GIV prototype (*Figure 1E*). In contrast, the lit conformation of LOV2GIVe displayed robust Gαi3 binding, thereby confirming an improved dynamic range compared to the LOV2GIV prototype (*Figure 1F*). Mutation of GIV's F1685 to alanine, a replacement known to preclude G-protein binding (*de Opakua et al., 2017*; *Garcia-Marcos et al., 2009*), resulted in greatly diminished binding of Gαi3 to the lit conformation of LOV2GIVe (*Figure 1—figure supplement 1*).

### LOV2GIVe binds and activates G-proteins efficiently in vitro only in its lit conformation

Concentration-dependent binding curves revealed that the affinity of Gαi3 for the dark conformation of LOV2GIVe is orders of magnitude weaker than for the lit conformation (*Figure 2A*), which had an equilibrium dissociation constant ($K_D$) similar to that previously reported for the GIV-Gαi3 interaction (*DiGiacomo et al., 2017*). We also found that LOV2GIVe retains the same G-protein specificity as GIV. Much like GIV (*Garcia-Marcos et al., 2010*; *Marivin et al., 2020*), LOV2GIVe bound robustly only to members of the $G_{i/o}$ family among the four families of Gα proteins ($G_{i/o}$, $G_s$, $G_{q/11}$, and $G_{12/13}$), and could discriminate within the $G_{i/o}$ family by binding to Gαi1, Gαi2 and Gαi3 but not to Gαo (*Figure 2B*). Next, we assessed if LOV2GIVe can activate G-proteins in vitro, that is whether it retains GIV's GEF activity. First, we measured Gαi3's steady-state GTPase activity in the presence of LOV2GIVe to assess its GEF activity. Under steady-state conditions, nucleotide exchange is rate limiting because GTP hydrolysis occurs orders of magnitude faster than GTP loading for Gαi proteins (*Mukhopadhyay and Ross, 2002*). Thus, steady-state GTP hydrolysis can report changes in the rate of nucleotide exchange on Gαi, which we have previously validated for GIV and other related non-receptor GEFs (*Aznar et al., 2015*; *Coleman et al., 2016*; *Garcia-Marcos et al., 2010*; *Garcia-Marcos et al., 2011*). We found that LOV2GIVe in the lit, but not the dark, conformation led to a dose-dependent increase of Gαi3 activity (*Figure 2C*), which was comparable to that previously shown for GIV (*Garcia-Marcos et al., 2009*). Using a GTPγS binding assay that measures nucleotide exchange more directly, we also found that LOV2GIVe in the lit conformation promoted nucleotide exchange on Gαi3, an effect that was impaired upon introduction of the F1685A (FA) mutation that

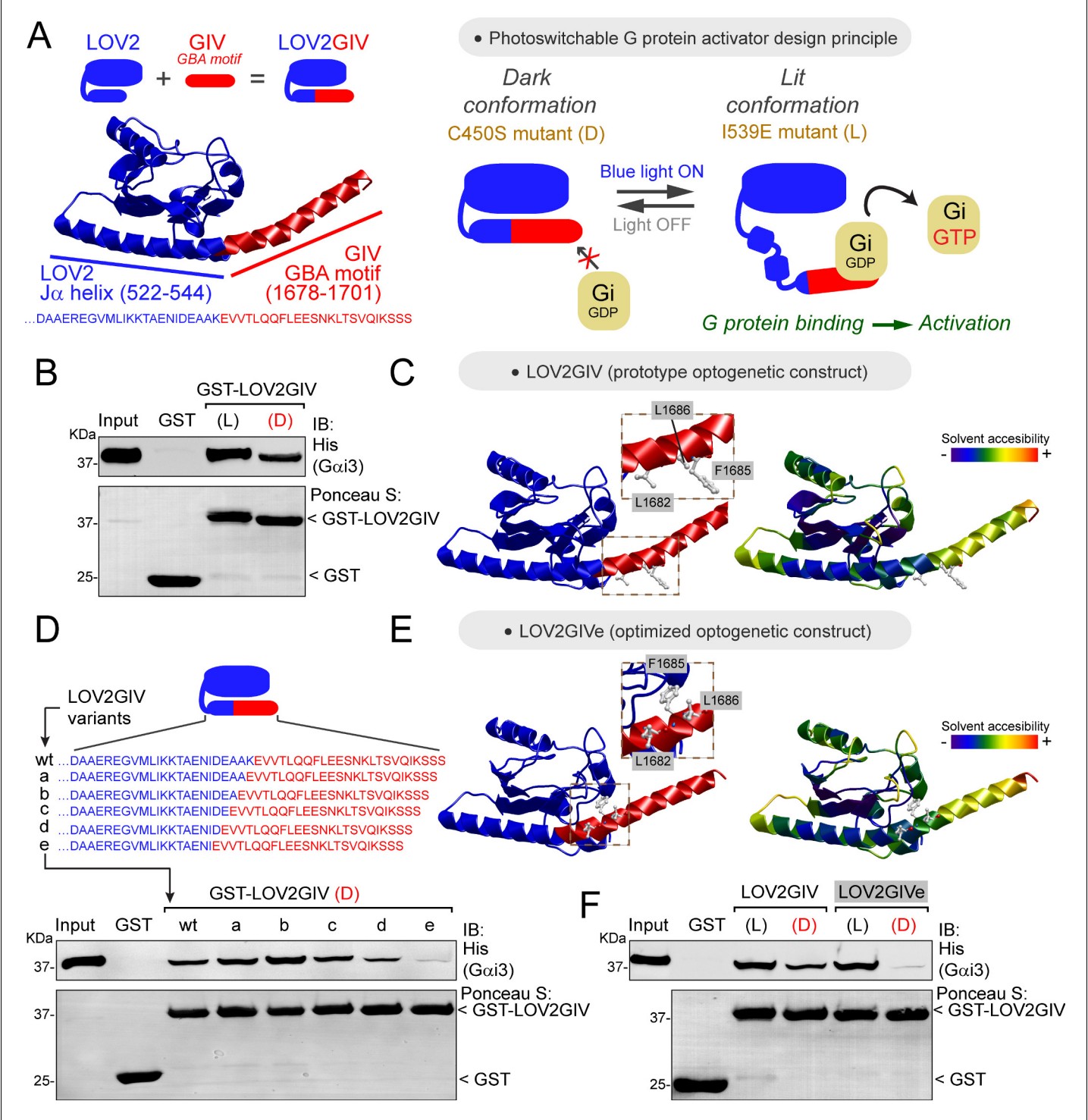

**Figure 1.** Design and optimization of LOV2GIV, an optogenetic activator of heterotrimeric G-proteins. (**A**) *Left,* diagram depicting the design of the LOV2GIV fusion protein. The GBA motif of GIV (red) is fused to the C-terminus of the LOV2 domain (blue). *Right,* diagram depicting the design principle of the photoswitchable G-protein activator LOV2GIV. In the dark conformation (D, which is mimicked by the LOV2 C450S mutant), the C-terminus of LOV2 forms an α-helix (Jα) and the GBA motif is not readily accessible for G-proteins. In the lit conformation (L, which is mimicked by the LOV2 mutant I539E), the C-terminus of LOV2 is more disordered and the GBA motif becomes accessible for G-proteins, which in turn are activated upon binding to the GBA motif. (**B**) LOV2GIV (L) binds Gαi3 better than LOV2GIV (D). Approximately 20 μg of the indicated purified GST-fused constructs were immobilized on glutathione-agarose beads and incubated with 3 μg (~300 nM) of purified His-Gαi3. Resin-bound proteins were eluted, separated by SDS-PAGE, and analyzed by Ponceau S-staining and immunoblotting (IB) with the indicated antibodies. Input = 10% of the total amount

*Figure 1 continued on next page*

*Figure 1 continued*

of His-Gαi3 added in each binding reaction. (**C**) Ribbon representation of a LOV2GIV structure homology model generated using the I-TASSER server. On the left, the model is colored blue for LOV2 and red for GIV, whereas on the right it is colored according to solvent accessibility. Selected GIV residues known to be important for G-protein binding (L1682, F1685, L1686) (*de Opakua et al., 2017*; *Kalogriopoulos et al., 2019*) are displayed in stick representation and enlarged in the boxes. (**D**) LOV2GIV (D) variant 'e' displays greatly diminished Gαi3 binding compared to LOV2GIV (D) wt. Protein binding experiments were carried out as described in panel B, except that the indicated LOV2GIV variants were used. (**E**) Ribbon representation of a structure homology model of the LOV2GIVe variant was generated and displayed as described for LOV2GIV in panel C. (**F**) The dynamic range of Gαi3 binding to lit versus dark conformations is improved for LOV2GIVe compared to LOV2GIV. Protein binding experiments were carried out as described in panel B, except that the indicated LOV2GIV variants were used. For all protein binding experiments in this figure, one representative result of at least three independent experiments is shown.

The online version of this article includes the following figure supplement(s) for figure 1:

**Figure supplement 1.** FA mutation in LOV2GIVe (L) impairs G-protein binding.

reduces G-protein binding or when using the dark-mimicking mutant (D) of LOV2GIVe (*Figure 2— figure supplement 1*). These findings indicate that LOV2GIVe recapitulates the G-protein activating properties of GIV in vitro, and that these are effectively suppressed for the dark conformation.

## LOV2GIVe activates G-protein signaling in cells in its lit conformation

To investigate LOV2GIVe-dependent G-protein activation in cells, we initially used a humanized yeast-based system (*Cismowski et al., 1999*; *DiGiacomo et al., 2020*) in which the mating pheromone response pathway has been co-opted to measure activation of human Gαi3 using a gene reporter (*LacZ*, β-galactosidase activity) (*Figure 3A*, left). When we expressed LOV2GIVe dark (D), lit (L) or wt in this strain, only very weak levels of β-galactosidase activity were detected (*Figure 3A*, right). We reasoned that this could be due to the subcellular localization of the construct, presumed to be cytosolic in the absence of any targeting sequence, because we have previously shown that GIV requires membrane localization to efficiently activate G-proteins (*Parag-Sharma et al., 2016*). Expressing LOV2GIVe (L) fused to a membrane-targeting sequence (mLOV2GIVe) led to a very strong induction of β-galactosidase activity, which was not recapitulated by expression of mLOV2GIVe (D) or wt (*Figure 3A*, right). LOV2GIVe-mediated activation levels (several hundred-fold over basal) are comparable to those previously reported for the endogenous pathway in response to GPCR activation by mating pheromone (*Hoffman et al., 2002*; *Lambert et al., 2010*). Next, we tested mLOV2GIVe in mammalian cells. Instead of using a downstream signaling readout as we did in yeast, we used a Bioluminescence Resonance Energy Transfer (BRET) biosensor that measures G-protein activity directly (*Figure 3B*, left). Expression of mLOV2GIVe (L), but not (D), led to an increase of BRET proportional to the amount of plasmid transfected (*Figure 3B*, right). At the highest amount tested, mLOV2GIVe (L) caused a BRET increase comparable to that observed upon maximal stimulation of the M4 muscarinic receptor, a Gi-activating GPCR (*Figure 3B*, right, *Figure 3— figure supplement 1*). Introducing a mutation that precludes G-protein activation by GIV's GBA motif (F1685A (FA), [*Garcia-Marcos et al., 2009*]), in mLOV2GIVe (L) also abolished its ability to induce a BRET increase (*Figure 3—figure supplement 2*). As an alternative readout of G-protein regulation by LOV2GIVe, we measured cAMP levels in cells because the enzymes that produce this second messenger, adenylyl cyclases, are classical effectors of GTP-bound Gαi subunits. Acute activation of Gi proteins leads to inhibition of adenylyl cyclase activity, but prolonged Gi activation leads to the opposite effect— it sensitizes adenylyl cyclases to subsequent activating inputs, such as the stimulation of a Gs-linked GPCR, leading to the potentiation of cAMP production (*Brust et al., 2015*; *Watts and Neve, 2005*). We found that expression of LOV2GIVe (L) leads to a potentiation of the cAMP response induced by the β-adrenergic agonist isoproterenol, suggesting prolonged activation of Gi proteins in cells (*Figure 3—figure supplement 3*). The magnitude of the potentiation (~2 fold) is comparable to that reported for many Gi-activating GPCRs (*Watts and Neve, 2005*). In contrast, expression of similar amounts of LOV2GIVe (L) bearing the F1685A (FA) mutation or of LOV2GIVe (D), which do not bind or activate Gαi, failed to recapitulate the potentiation of the cAMP response elicited by LOV2GIV (L) (*Figure 3—figure supplement 3*), further supporting the involvement of Gαi activation. Taken together, our results indicate that the lit conformation of LOV2GIVe activates G-protein signaling in different cell types.

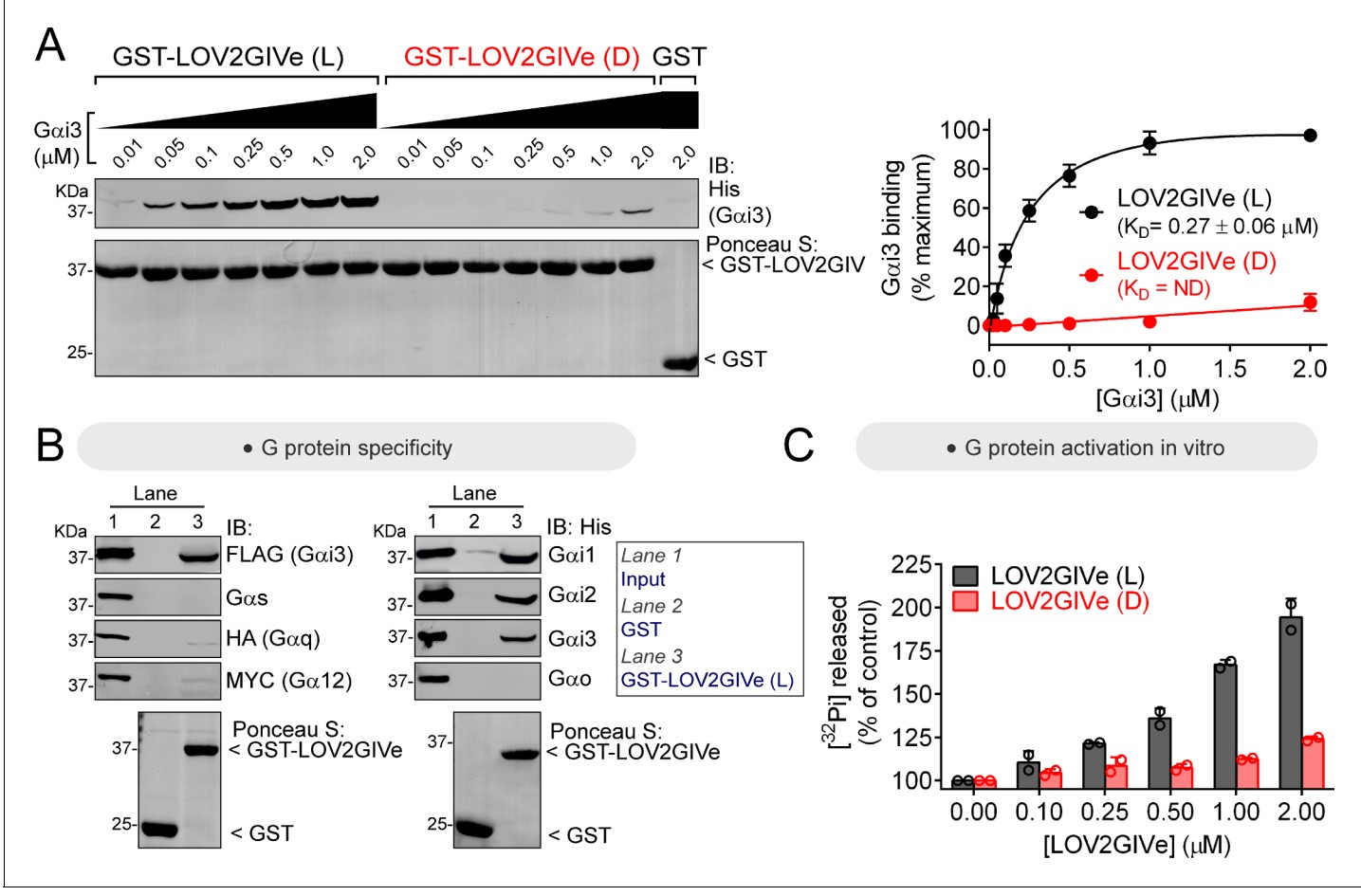

**Figure 2.** LOV2GIVe binds and activates Gαi3 in vitro in its lit conformation. (**A**) LOV2GIVe binds with high affinity to Gαi3. Approximately 10 µg of the indicated purified GST-fused constructs were immobilized on glutathione-agarose beads and incubated with the indicated concentrations of purified His-Gαi3. Resin-bound proteins were eluted, separated by SDS-PAGE, and analyzed by Ponceau S-staining and immunoblotting (IB) with the indicated antibodies. One representative result is shown on the left, and the graph on the right corresponds to the quantification of three independent experiments presented as mean ± S.E.M. for each data point and a solid line for the fit to a single site binding curve used to determine the $K_D$ values. (**B**) LOV2GIVe binds specifically to Gαi compared to other Gα subunits. Approximately 20 µg of the indicated purified GST-fused constructs were immobilized on glutathione-agarose beads and incubated with the lysates of HEK293T cells expressing the indicated G-proteins (FLAG-Gαi3, Gαs, Gαq-HA and Gα12-MYC on the left panels) or purified His-tagged proteins (3 µg, ~300 nM of His-Gαi1, His-Gαi2, His-Gαi3 and His-Gαo on the right panels). One representative experiment of at least three is shown. (**C**) LOV2GIVe (L), but not LOV2GIVe (D), increases Gαi3 activity in vitro. Steady-state GTPase activity of purified His-Gαi3 was determined in the presence of increasing amounts (0–2 µM) of purified GST-LOV2GIVe (L) (black) or GST-LOV2GIVe (D) (red) by measuring the production of [$^{32}$P]Pi at 15 min. Results are the mean ± S.D. of n = 2.

The online version of this article includes the following source data and figure supplement(s) for figure 2:

**Source data 1.** Numerical data used for panel A.

**Source data 2.** Numerical data used for panel C.

**Figure supplement 1.** LOV2GIVe (L) promotes GTP binding to Gαi3 in vitro.

**Figure supplement 1—source data 1.** Numerical data used to generate the graph.

## LOV2GIVe activates G-protein signaling in cells upon illumination

Next, we tested whether LOV2GIVe can trigger G-protein activation in cells in response to light. For this, we used two complementary experimental systems. The first one consisted of measuring G-protein activity directly with the mammalian cell BRET biosensor described above upon illumination with a pulse of blue light. We found that, in HEK293T cells expressing mLOV2GIVe wt, a single short light pulse resulted in a spike of G-protein activation that decayed at a rate similar to that reported for the transition from lit to dark conformation of LOV2 ($T_{1/2}$ ~1 min) (*Figure 4A*). This response was not recapitulated by a mLOV2GIVe construct bearing the GEF-deficient FA mutation (*Figure 4A*). For

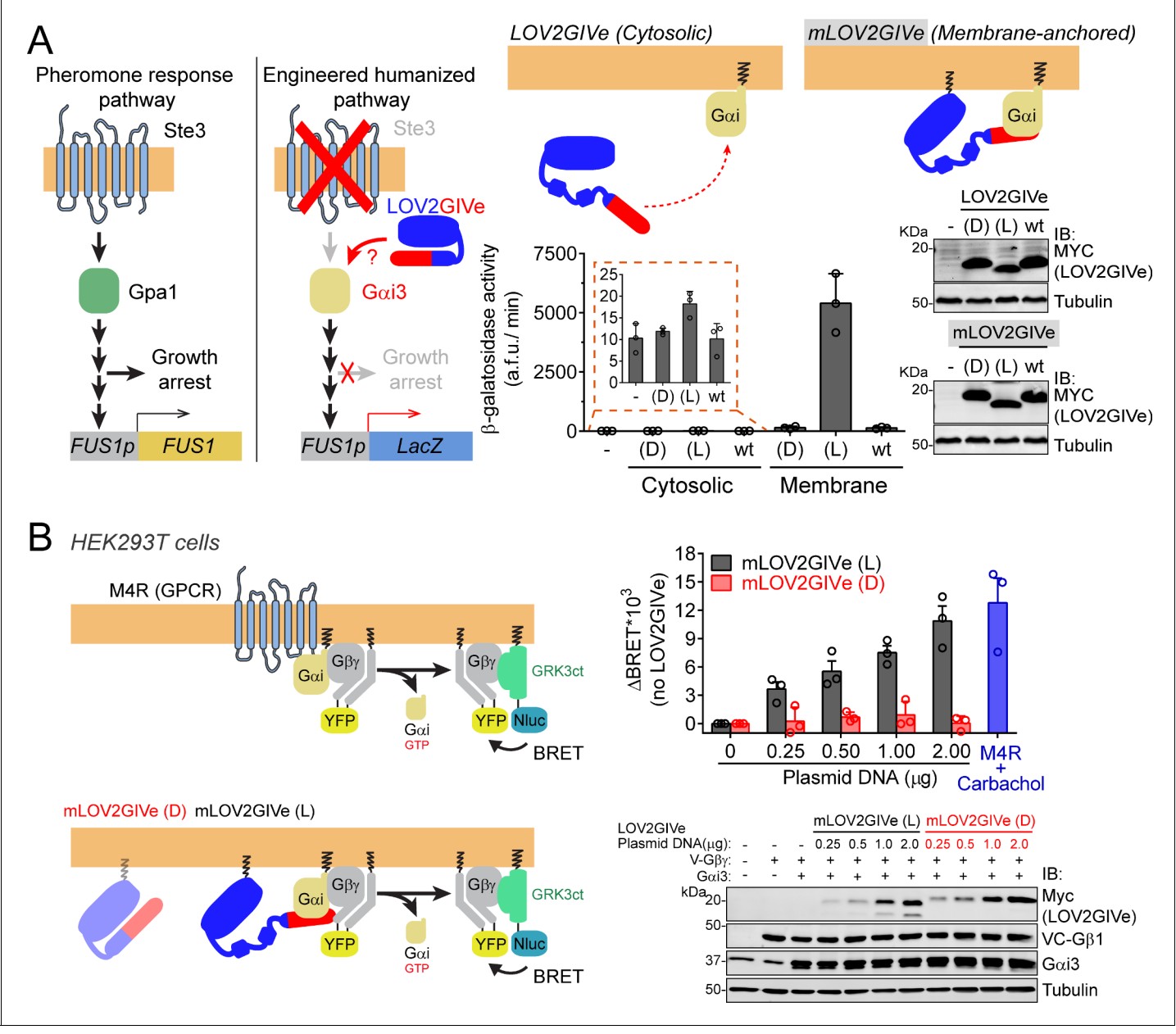

**Figure 3.** LOV2GIVe activates G-protein signaling in cells in its lit conformation. (**A**) *Left*, diagram comparing key steps and components of the mating pheromone response pathway of *Saccharomyces cerevisiae* to those of an engineered, humanized strain used in the experiments shown in this figure. In the engineered strain, no pheromone responsive GPCR (Ste3) is expressed, the yeast G-protein Gpa1 is replaced by human Gαi3, and downstream signaling does not lead to growth arrest but promotes the activation of a reporter gene (*LacZ*) under the control of the pheromone-sensitive, G-protein-dependent promoter of *FUS1* (*FUS1p*). *Right*, membrane-anchored LOV2GIVe (mLOV2GIVe), but not its untargeted parental version, leads to strong G-protein activation only in the lit conformation. Yeast strains expressing the indicated LOV2GIVe constructs ((D), (L) or wt) or an empty vector (-) were lysed to determine β-galactosidase activity using a fluorogenic substrate (mean ± S.E.M., n = 3) and to prepare samples for immunoblotting (IB)(one experiment representative of 3 is shown). (**B**) *Left*, diagrams showing the principle for the G-protein activity biosensor used in this panel. Upon action of a GPCR (top) or mLOV2GIVe (bottom), G-protein activation leads to the release of YFP-tagged Gβγ from Gαi, which then can associate with Nluc-tagged GRK3ct and results in the subsequent increase in BRET. *Right*, mLOV2GIVe (L) but not mLOV2GIVe (D), leads to increased G-protein activation similar to a GPCR as determined by BRET. BRET was measured in HEK293T cells transfected with the indicated amounts of mLOV2GIVe plasmid constructs along with the components of the BRET biosensor and the GPCR M4R. M4R was stimulated with 100 μM carbachol. Results in the graph on the top are expressed as difference in BRET (ΔBRET) relative to unstimulated cells not expressing mLOV2GIVe (mean ± S.E.M., n = 3). One representative immunoblot from cell lysates made after the BRET measurements is shown on the bottom.

The online version of this article includes the following source data and figure supplement(s) for figure 3:

**Source data 1.** Numerical data used for panel A.

*Figure 3 continued on next page*

*Figure 3 continued*

**Source data 2.** Numerical data used for panel B.
**Figure supplement 1.** Carbachol dose-dependent G-protein activation and its blockade by atropine.
**Figure supplement 1—source data 1.** Numerical data used to generate the graph.
**Figure supplement 2.** FA mutation abolishes G-protein activation by mLOV2GIVe (L) in cells.
**Figure supplement 2—source data 1.** Numerical data used to generate the graph.
**Figure supplement 3.** LOV2GIVe (L) expression enhances isoproterenol-induced cAMP levels in cells via Gi regulation.
**Figure supplement 3—source data 1.** Numerical data used for panel A.
**Figure supplement 3—source data 2.** Numerical data used for panel B.

the second system, we turned to the humanized yeast strain described above. Instead of using β-galactosidase assays to report G-protein dependent activation of the *FUS1* promoter by light, we assayed conditional histidine prototrophy controlled by the *FUS1* promoter using spot growth assays because they are better suited to allow homogenous and continued illumination than the cell suspension conditions used of the β-galactosidase assay (*Figure 4B*, left). Yeast cells expressing mLOV2GIVe wt grew in the absence of histidine only when exposed to blue light (*Figure 4B*, right). This effect was specifically caused by light-dependent activation of mLOV2GIVe wt because cells expressing the light-insensitive mLOV2GIVe (D) construct failed to grow under the same illumination conditions, whereas mLOV2GIVe (L) grew the same regardless of illumination conditions (*Figure 4B*, right). Taken together, these results show that mLOV2GIVe activates G-protein signaling in cells upon blue light illumination.

## Conclusions and future perspectives

Here, we have presented proof-of-principle evidence for a photoswitchable G-protein activator that does not rely on opsins, that is light-activated GPCRs. This tool is based on a modular design that combines the properties of the light-sensitive LOV2 domain with a motif present in a non-GPCR activator of G-proteins of the Gi family. We propose that the versatility of the LOV2GIVe design could help overcome some limitations of currently available optogenetic tools that are based on GPCR-like proteins. For example, recent evidence suggests that opiod receptors can activate Gi proteins in different subcellular compartments (*Stoeber et al., 2018*), but it has not been possible to control GPCR activation in specific subcellular compartments to address the consequences of spatially encoded signals. Given that our results indicate that LOV2GIVe requires targeting to membranes where the substrate G-protein localizes (*Figure 3A*), it would be possible in the future to target it to different membranous organelles to trigger Gi activation in specific subcellular compartments. LOV2GIVe could also be useful for neurobiogical applications because Gi proteins are critical mediator of inhibitory neuromodulation. Combined with the potential for subcellular compartmentalization, LOV2GIVe could be leveraged to assess the different impact of inhibitory modulation at pre- or post-synaptic sites. Our results also support that LOV2GIVe can activate G-protein signaling even in cellular systems where there is not sufficient synthesis of retinal to support opsin-based activation, like in the yeast *S. cerevisiae* (*Scott et al., 2019*). This could open the application of optogenetic Gi activation to organisms or experimental settings in which the lack of external supplementation of retinal is detrimental, such those required prolonged and/or repeated stimulation. A general advantage of optogenetic approaches over other synthetic biology approaches to activate G-proteins like chemogenetics with Designer Receptors Exclusively Activated by Designer Drugs (DREADDs) (*Urban and Roth, 2015*) is that they allow for the rapid and accurate temporal control of both activation and deactivation. Thus, LOV2GIVe could be used to investigate Gi signaling processes over a broad temporal scale, also including intermittent or pulsatile activation. A limitation of LOV2GIVe is that it only acts on one subset of heterotrimeric G-proteins, those containing Gαi subunits. Nevertheless, potential applications are still broad, as Gi proteins control processes as diverse as inhibitory neuromodulation, opioid action, or heart rate, among many others.

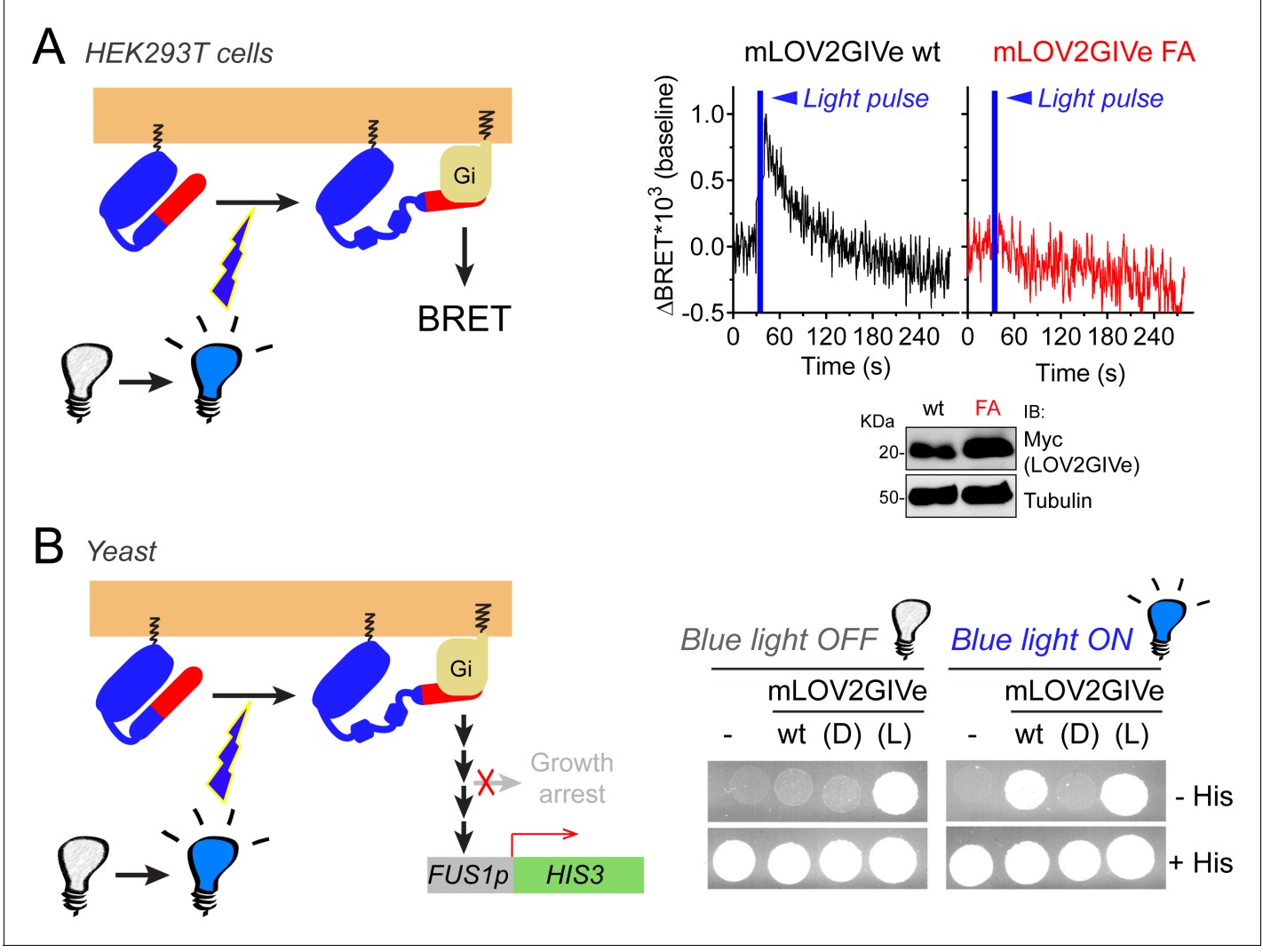

**Figure 4.** mLOV2GIVe activates G-protein signaling in cells upon illumination. (**A**) mLOV2GIVe activates G-proteins in mammalian cells upon illumination. Kinetic BRET measurements were carried out in HEK293T cells transfected with 2 µg of plasmids for the expression of mLOV2GIVe wt or FA along with the components of the BRET biosensor. Results are expressed as baseline corrected BRET changes (ΔBRET) and representative traces of one experiment out of four are presented. (**B**) mLOV2GIVe activates G-protein signaling in cells upon blue light illumination. Yeast strains expressing the indicated mLOV2GIVe constructs (wt, (D) or (L)) or an empty vector (-) were spotted on plates with or without histidine as indicated and imaged after 4 days of incubation in the dark (blue light OFF) or in the presence of blue light (blue light ON). One experiment representative of 3 is shown. The online version of this article includes the following source data for figure 4:

**Source data 1.** Numerical data used in panel A.

## Materials and methods

### Reagents and antibodies

Unless otherwise indicated, all chemical reagents were obtained from Sigma or Fisher Scientific. Fluorescein di-β-D-galactopyranoside (FDG) was from Marker Gene Technologies, and the protein inhibitor mixture was from Sigma (catalog no. S8830). Leupeptin, pepstatin, and aprotinin were from Gold Biotechnology. All restriction endonucleases and *E. coli* strain BL21(DE3) were from Thermo Scientific. *E. coli* strain DH5α was purchased from New England Biolabs. *Pfu* ultra DNA polymerase used for site-directed mutagenesis was purchased from Agilent. [γ-$^{32}$P]GTP was from Perkin Elmer. Mouse monoclonal antibodies raised against α-tubulin (T6074), FLAG tag (F1804) or His-tag (H1029) were from Sigma. Mouse monoclonal antibody raised against hemagglutinin (HA) tag (clone 12CA5, #11583816001) was obtained from Roche. Mouse monoclonal antibody raised against MYC-tag

(9B11, #2276) was from Cell Signaling. Rabbit polyclonal antibody raised against Gαs (C-18, sc-383) was purchased from Santa Cruz Biotechnology. Goat anti-rabbit Alexa Fluor 680 (A21077) and goat anti-mouse IRDye 800 (#926–32210) secondary antibodies were from Life technologies and LiCor, respectively.

## Plasmid constructs

The parental LOV2GIV sequence was obtained as a synthetic gene fragment from GenScript and subsequently amplified by PCR with extensions at the 5' and 3' ends that made it compatible with a ligation-independent cloning (LIC) system (*Stols et al., 2002*). For the bacterial expression of GST-LOV2GIV constructs, we inserted the LOV2GIV sequence into the pLIC-GST plasmid kindly provided by J. Sondek (UNC-Chapel Hill) (*Cabrita et al., 2006*) to generate the plasmid pLIC-GST-LOV2GIV. Two previously described LOV2 mutations (L514K and L531E) that reduce the spurious unwinding of the Jα helix in the dark conformation (*Lungu et al., 2012*; *Zimmerman et al., 2016*) were introduced in this construct. These and other mutations to generate the dark (D, C450S) and lit (L, I539E) confor-mations, the a, b, c, d, and e variants described on *Figure 1D*, and the GEF-deficient FA mutant were made using the QuikChange II Mutagenesis Kit from Agilent. Full sequences of the parental LOV2GIV and LOV2GIVe variant are provided in Supplementary Information. For the yeast expres-sion of LOV2GIVe constructs, we inserted the LOV2GIVe sequence into two different versions of a previously described pYES2 derived plasmid (*Coleman et al., 2016*): pLIC-YES2 and pLIC-YES2-N9Gpa1. Both versions contain a MYC-tag sequence cloned upstream of the LIC cassette between the HindIII and KpnI sites, but in one of the two plasmids it was preceded by a sequence encoding the first nine amino acids of *S. cerevisiae* Gpa1, a previously validated membrane-targeting sequence (*Parag-Sharma et al., 2016*; *Song et al., 1996*). For the mammalian expression of LOV2-GIVe constructs, we inserted the LOV2GIVe sequence into a modified pLIC-myc plasmid (*Cabrita et al., 2006*, kindly provided by J. Sondek (UNC-Chapel Hill, NC)), in which a sequence encoding the first 11 amino acids of Lyn, a previously validated membrane-targeting sequence (*Inoue et al., 2005*; *Parag-Sharma et al., 2016*), was inserted in the AflII/KpnI sites upstream of the MYC-tag (pLIC-lyn11-myc). The resulting sequence preceding the first amino acid of LOV2GIVe is **MGCIKSKGKD**SGTELGSM<u>EQKLISEEDL</u>GILYFQSNA (bold = Lyn11, underline = MYC-tag). Cloning of the pET28b-Gαi3 and pET28b-Gαo plasmids for the bacterial expression of rat His-Gαi3 or rat His-Gαo (isoform A), respectively, have been described previously (*Garcia-Marcos et al., 2010*; *Gar-cia-Marcos et al., 2009*). Plasmid pLIC-Gαi1(int.6xHis) for the bacterial expression of rat Gαi1 with an internal hexahistidine tag at position 120 was generated by PCR amplification from pQE6-Gαi1 (int.6xHis) (*Dessauer et al., 1998*, kindly provided by Carmen Dessauer, University of Texas, Hous-ton) and insertion at NdeI/BglII sites of the pLIC-His plasmid (*Stols et al., 2002*). The plasmid for the bacterial expression of rat His-Gαi2 (pET28b-Gαi2) was generated by inserting the Gαi2 sequence into the NdeI/EcoRI of pET28b. Plasmids for expression of FLAG-Gαi3 (rat, p3XFLAG-CMV10-Gαi3, N-terminal 3XFLAG tag) or Gαs (human, pcDNA3.1(+)-Gαs) in mammalian cells were described pre-viously (*Beas et al., 2012*; *Garcia-Marcos et al., 2009*). Plasmids for the expression of Gαq-HA (mouse, pcDNA3-Gαq-HA, internally tagged) or Gα12-MYC (mouse, pcDNA3.1-Gα12-MYC, inter-nally tagged) in mammalian cells were kindly provided by P. Wedegaertner (Thomas Jefferson Uni-versity) (*Wedegaertner et al., 1993*) and T. Meigs (University of North Carolina, Asheville) (*Ritchie et al., 2013*), respectively. Plasmid for the expression of Gβ1 and Gγ2 fused to a split Venus (pcDNA3.1-Venus(1-155)-Gγ$_2$ (VN-Gγ$_2$) and pcDNA3.1-Venus(155-239)-Gβ$_1$ (VC-Gβ$_1$)) or for untagged Gβ1 (pcDNA3.1-Gβ$_1$) and Gγ2 (pcDNA3.1-Gγ$_2$) were kindly provided by N. Lambert (Augusta University, GA) (*Hollins et al., 2009*). pcDNA3.1-masGRK3ct-Nluc and pcDNA3.1-Nluc-EPAC-VV (*Masuho et al., 2015*) were a gift from K. Martemyanov (Scripps Research Institute, FL). The pcDNA3-Gαi3 plasmid for the expression of rat Gαi3 in mammalian cells has been described previously (*Garcia-Marcos et al., 2010*; *Garcia-Marcos et al., 2009*). The plasmid for the expression of M4R was obtained from the cDNA Resource Center at Bloomsburg University (pcDNA3.1-3xHA-M4R, cat# MAR040TN00).

## Protein expression and purification

All His-tagged and GST-tagged proteins were expressed in BL21(DE3) *E. coli* transformed with the corresponding plasmids by overnight induction at 23°C with 1 mM isopropyl-β-D-1-thio-

galactopyranoside (IPTG). Protein purification was carried out following previously described protocols (*Garcia-Marcos et al., 2010*; *Garcia-Marcos et al., 2009*). Briefly, bacteria pelleted from 1 L of culture were resuspended in 25 mL of buffer [50 mM $NaH_2PO_4$, pH 7.4, 300 mM NaCl, 10 mM imidazole, 1% (v:v) Triton X-100 supplemented with protease inhibitor cocktail (Leupeptin 1 µM, Pepstatin 2.5 µM, Aprotinin 0.2 µM, PMSF 1 mM)]. For His-Gαi3 and His-Gαo, this buffer was supplemented with 25 µM GDP and 5 mM $MgCl_2$. After sonication (four cycles, with pulses lasting 20 s/cycle, and with 1 min interval between cycles to prevent heating), lysates were centrifuged at 12,000 g for 20 min at 4°C. The soluble fraction (supernatant) of the lysate was used for affinity purification on HisPur cobalt or glutathione- agarose resins (Pierce) and eluted with lysis buffer supplemented with 250 mM imidazole or with 50 mM Tris-HCl, pH 8, 100 mM NaCl, 30 mM reduced glutathione, respectively. GST-tagged proteins were dialyzed overnight at 4°C against PBS. For His-Gαi1, His-Gαi2, His-Gαi3 and His-Gαo, the buffer was exchanged for 20 mM Tris-HCl, pH 7.4, 20 mM NaCl, 1 mM $MgCl_2$, 1 mM DTT, 10 µM GDP, 5% (v/v) glycerol using a HiTrap Desalting column (GE Healthcare). All protein samples were aliquoted and stored at −80°C.

## Protein–protein binding assays

GST pulldown assays were carried out as described previously (*Garcia-Marcos et al., 2010*; *Garcia-Marcos et al., 2011*) with minor modifications. GST or GST-fused LOV2GIV constructs (described in '*Plasmid Constructs*') were immobilized on glutathione-agarose beads for 90 min at room temperature in PBS. Beads were washed twice with PBS, resuspended in 250–400 µL of binding buffer (50 mM Tris-HCl, pH 7.4, 100 mM NaCl, 0.4% (v:v) NP-40, 10 mM $MgCl_2$, 5 mM EDTA, 1 mM DTT, 30 µM GDP) and incubated 4 hr at 4°C with constant tumbling in the presence of His-tagged G-proteins purified as described in '*Protein Expression and Purification*' or lysates of HEK293T cells expressing different G-proteins. For the latter, HEK293T cells (ATCC CRL3216) were grown at 37°C, 5%$CO_2$ in high-glucose Dulbecco's Modified Eagle Medium (DMEM) supplemented with 10% FBS, 100 U/mL penicillin, 100 µg/mL streptomycin and 1% L-glutamine. HEK293T cells were not authenticated by STR profiling or tested for mycoplasma contamination. Approximately two million HEK293T cells were seeded on 10 cm dishes and transfected the day after using the calcium phosphate method with plasmids encoding the following constructs (DNA amounts in parenthesis): FLAG-Gαi3 (3 µg), Gαs (3 µg), Gαq-HA (6 µg) or Gα12-MYC (3 µg). Cell medium was changed 6 hr after transfection. Thirty two hours after transfection, cells were harvested by scraping in PBS and centrifugation before resuspension in 500 µL of ice-cold lysis buffer (20 mM Hepes, pH 7.2, 125 mM K($CH_3$COO), 0.4% (vol:vol) Triton X-100, 1 mM DTT, 10 mM β-glycerophosphate and 0.5 mM $Na_3VO_4$ supplemented with a protease inhibitor cocktail [SigmaFAST, #S8830]). Cell lysates were cleared by centrifugation at 14,000 g for 10 min at 4°C. Approximately, 20–25% of the lysate from a 10 cm dish was used for each binding reaction. After incubation with purified proteins or cell lysates, beads were washed four times with 1 mL of wash buffer (4.3 mM $Na_2HPO_4$, 1.4 mM $KH_2PO_4$, pH 7.4, 137 mM NaCl, 2.7 mM KCl, 0.1% (v/v) Tween-20, 10 mM $MgCl_2$, 5 mM EDTA, 1 mM DTT, and 30 µM GDP) and resin-bound proteins eluted by boiling for 5 min in Laemmli sample buffer before processing for IB (see below '*Immunoblotting*').

## Protein structure modeling and visualization

Models of LOV2GIV and LOV2GIVe were generated using the server I-TASSER (https://zhanglab.ccmb.med.umich.edu/I-TASSER/, [*Yang et al., 2015*]). Best scoring models were chosen for further analysis (LOV2GIV C-score = −0.64, LOV2GIVe C-score = −0.33). Protein structures were visualized and displayed using ICM version 3.8–3 (Molsoft LLC., San Diego, CA).

## Steady-state GTPase assay

This assay was performed as described previously (*Garcia-Marcos et al., 2010*; *Garcia-Marcos et al., 2009*; *Garcia-Marcos et al., 2011*). Briefly, His-Gαi3 (400 nM) was pre-incubated with different concentrations of GST-LOV2GIVe constructs for 15 min at 30°C in assay buffer [20 mM Na-HEPES, pH 8, 100 mM NaCl, 1 mM EDTA, 25 mM $MgCl_2$, 1 mM DTT, 0.05% (w:v) $C_{12}E_{10}$]. GTPase reactions were initiated at 30°C by adding an equal volume of assay buffer containing 1 µM [$γ$-$^{32}$P] GTP (~50 c.p.m/ fmol). Duplicate aliquots (25 µL) were removed at 15 min and reactions stopped with 975 µL of ice-cold 5% (w/v) activated charcoal in 20 mM $H_3PO_4$, pH 3. Samples were then

centrifuged for 10 min at 10,000 g, and 500 µL of the resultant supernatant were scintillation counted to quantify [$^{32}$P]Pi released. Background [$^{32}$P]Pi detected at 15 min in the absence of G-protein was subtracted from each reaction and data expressed as percentage of the Pi produced by His-Gαi3 in the absence of GST-LOV2GIVe. Background counts were <5% of the counts detected in the presence of G-proteins.

## GTPγs binding assay

GTPγS binding to purified His-Gαi3 was determined as described previously (*Garcia-Marcos et al., 2010*; *Leyme et al., 2014*). Purified His-Gαi3 (100 nM) was diluted in assay buffer (20 mM Na-HEPES, pH 8, 100 mM NaCl, 1 mM EDTA, 25 mM MgCl$_2$, 1 mM DTT, 0.05% (wt:vol) C$_{12}$E$_{10}$) and pre-incubated with purified GST-LOV2GIVe proteins (2 µM final) for 15 min at 30°C. Reactions were initiated, by adding an equal volume of assay buffer containing 1 µM [$^{35}$S]GTPγS (~50 c.p.m/ fmol) at 30°C. Duplicate aliquots (25 µL) were removed 15 min after the reaction start, and binding of radio-active nucleotide was stopped by addition of 2 mL of ice-cold wash buffer (20 mM Tris-HCl, pH 8.0, 100 mM NaCl, 25 mM MgCl$_2$). The quenched reactions were rapidly passed through BA-85 nitrocel-lulose filters (GE Healthcare) and washed with 2 mL cold wash buffer. Filters were dried and sub-jected to liquid scintillation counting. Background [$^{35}$S]GTPγS detected in the absence of G-protein was subtracted from each reaction and data expressed as percentage of the [$^{35}$S]GTPγS bound to His-Gαi3 in the absence of GST-LOV2GIVe.

## Yeast strains and manipulations

The previously described (*Cismowski et al., 1999*) *S. cerevisiae* strain CY7967 [*MATα GPA1(1–41)-Gαi3 far1Δ fus1p-HIS3 can1 ste14:trp1:LYS2 ste3Δ lys2 ura3 leu2 trp1 his3*] (kindly provided by James Broach, Penn State University) was used for all yeast experiments. The main features of this strain are that the gene encoding only pheromone responsive GPCR (*STE3*) is deleted, the endoge-nous Gα-subunit Gpa1 is replaced by a chimeric Gpa1(1-41)-human Gαi3 (36-354) and the gene encoding the cell cycle arrest-inducing protein Far1 is deleted. In this strain, the pheromone response pathway can be upregulated by the ectopic expression of activators of human Gαi3 and does not result in the cell cycle arrest that occurs in the native pheromone response (*Cismowski et al., 2002*; *Cismowski et al., 1999*; *Maziarz et al., 2018*). Plasmid transformations were carried out using the lithium acetate method. CY7967 was first transformed with a centromeric plasmid (CEN TRP) encoding the *LacZ* gene under the control of the *FUS1* promoter (*FUS1p*), which is activated by the pheromone response pathway. The *FUS1p::LacZ*-expressing strain was trans-formed with pLIC-YES2 plasmids (2 µm, URA) encoding each of the LOV2GIV constructs described in '*Plasmid Constructs*'. Double transformants were selected in synthetic defined (SD)-TRP-URA media. Individual colonies were inoculated into 3 mL of SDGalactose-TRP-URA and incubated over-night at 30°C to induce the expression of the proteins of interest under the control of the galactose-inducible promoter of pLIC-YES2. This starting culture was used to inoculate 20 mL of SDGalactose-TRP-URA at 0.3 OD600. Exponentially growing cells (~0.7–0.8 OD600, 4–5 hr) were pelleted to pre-pare samples for '*β-galactosidase Activity Assay*' and '*Yeast Spot Growth Assay*' described below and for preparing samples for immunobloting as previously described (*de Opakua et al., 2017*; *Maziarz et al., 2018*). Briefly, pellets corresponding to 5 OD600 were washed once with PBS + 0.1% BSA and resuspended in 150 µL of lysis buffer (10 mM Tris-HCl, pH 8.0, 10% (w:v) trichloroacetic acid (TCA), 25 mM NH$_4$OAc, 1 mM EDTA). 100 µL of glass beads were added to each tube and vor-texed at 4°C for 5 min. Lysates were separated from glass beads by poking a hole in the bottom of the tubes followed by centrifugation onto a new set of tubes. The process was repeated after the addition of 50 µL of lysis buffer to wash the glass beads. Proteins were precipitated by centrifugation (10 min, 20,000 g) and resuspended in 60 µL of solubilization buffer (0.1 M Tris-HCl, pH 11.0, 3% SDS). Samples were boiled for 5 min, centrifuged (1 min, 20,000 g), and 50 µL of the supernatant transferred to new tubes containing 12.5 µL of Laemmli sample buffer and boiled for 5 min.

## β-galactosidase activity assay

This assay was performed as described previously (*Hoffman et al., 2002*; *Maziarz et al., 2018*) with minor modifications. Pellets corresponding to 0.5 OD600 (in duplicates) were washed once with PBS + 0.1% (w:v) BSA and resuspended in 200 µL assay buffer (60 mM Na$_2$PO$_4$, 40 mM NaH$_2$PO$_4$, 10

mM KCl, 1 mM MgCl$_2$, 0.25% (v:v) β-mercaptoethanol, 0.01% (w:v) SDS, 10% (v:v) chloroform) and vortexed. 100 μL were transferred to 96-well plates and reactions started by the addition of 50 μL of the fluorogenic β-galactosidase substrate fluorescein di-β-D-galactopyranoside (FDG, 100 μM final). Fluorescence (Ex. 485 ± 10 nm/ Em. 528 ± 10 nm) was measured every 2 min for 90 min at 30°C in a Biotek H1 synergy plate reader. Enzymatic activity was calculated as arbitrary fluorescent units (a.f. u.) per minute (min).

## BRET-based G-protein activation assay

BRET experiments were conducted as described previously (*Maziarz et al., 2018*). HEK293T cells (ATCC, CRL-3216) were seeded on 6-well plates (~400,000 cells/well) coated with gelatin and after one day transfected using the calcium phosphate method with plasmids encoding for the following constructs (DNA amounts in parenthesis): Venus(155-239)-Gβ$_1$ (VC-Gβ$_1$) (0.2 μg), Venus(1-155)-Gγ$_2$ (VN-Gγ$_2$) (0.2 μg) and Gαi3 (1 μg) mas-GRK3ct-Nluc (0.2 μg) along with mLOV2GIVe constructs in the amounts indicated in the corresponding figure legends. Approximately 16–24 hr after transfection, cells were washed and gently scraped in room temperature PBS, centrifuged (5 min at 550 g) and resuspended in assay buffer (140 mM NaCl, 5 mM KCl, 1 mM MgCl$_2$, 1 mM CaCl$_2$, 0.37 mM NaH$_2$PO$_4$, 20 mM HEPES pH 7.4, 0.1% glucose) at a concentration of 1 million cells/mL. 25,000–50,000 cells were added to a white opaque 96-well plate (Opti-Plate, Perkin Elmer) and mixed with the nanoluciferase substrate Nano-Glo (Promega cat# N1120, final dilution 1:200) for 2 min before measuring luminescence signals in a POLARstar OMEGA plate reader (BMG Labtech) at 28°C. Luminescence was measured at 460 ± 40 nm and 535 ± 10 nm, and BRET was calculated as the ratio between the emission intensity at 535 ± 10 nm divided by the emission intensity at 460 ± 40 nm. For the activation of M4R in *Figure 3B*, cells were exposed to 100 μM carbachol for 4 min prior to measuring BRET. For measurements shown in *Figure 3B* and *Figure 4B*, BRET data are presented as the difference from cells not expressing LOV2GIVe constructs. For kinetic BRET measurements shown in *Figure 4A*, luminescence signals were measured every 0.24 s for the duration of the experiment. The illumination pulse was achieved by switching from the luminescence read mode to the fluorescence read mode of the plate reader, with the following settings: 485 ± 6 nm filter, 200 flashes (~1.5 s). After the pulse of illumination, measurements were returned to luminescence mode with the same settings as prior to illumination. BRET data were corrected by subtracting the BRET signal baseline (average of 30 s pre-light pulse) and then subjected to a smoothening function (second order, four neighbors) in GraphPad for presentation. At the end of some BRET experiments, a separate aliquot of the same pool of cells used for the luminescence measurements was centrifuged for 1 min at 14,000 g and pellets stored at −20°C. To prepare lysates for IB, pellets were resuspended in lysis buffer (20 mM HEPES, pH 7.2, 5 mM Mg(CH3COO)$_2$, 125 mM K(CH3COO), 0.4% Triton X-100, 1 mM DTT, and protease inhibitor mixture). After clearing by centrifugation at 14,000 g at 4°C for 10 min, protein concentration was determined by Bradford and samples boiled in Laemmli sample buffer for 5 min before following the procedures described in '*Immunoblotting*'.

## Intracellular cAMP measurements

This assay was performed using the previously described BRET-based biosensor NLuc-EPAC-VV (*Leyme et al., 2017*; *Masuho et al., 2015*). HEK293T cells were seeded, transfected, and harvested as described above ('*BRET-based G-protein activation assay*') except that the plasmids used were (quantities in parenthesis): Nluc-EPAC-VV (0.05 μg), Gαi3 (1 μg) Gβ1 (0.5 μg), Gγ2 (0.5 μg) and the mLOV2GIVe constructs indicated in the figures (2 μg). A POLARstar OMEGA plate reader (BMG Labtech) was used to measure luminescence signals at 460 ± 20 nm and 528 ± 10 nm at 28°C every 5 s and BRET calculated as the ratio between the emission intensity at 528 ± 10 nm divided by the emission intensity at 460 ± 20 nm. Results were normalized to the basal BRET ratio before the addition of isoproterenol and presented as the inverse of this normalized BRET ratio. Isoproterenol (0.1 μM) was added at 60 s. Preparation of protein samples for IB was performed as described in '*BRET-based G-Protein Activation Assay*'.

## Yeast spot growth assay

Cells bearing LOV2GIVe constructs growing exponentially in SDGalactose media were pelleted as described above ('*Yeast Strains and Manipulations*'), and resuspended at equal densities. Equal

volumes of each strain were spotted on agar plates in four identical sets. Two of the sets were seeded on SDGalactose-TRP-URA plates with histidine and the other two sets were seeded on SDGalactose-TRP-URA-HIS (supplemented with 5 mM 3-amino-1,2,4-triazole). From each one of the two pairs of sets, one of the plates was exposed to a homemade array of blue LED strips positioned approximately 12 cm above the plates (~2,000 Lux as determined by a Trendbox Digital Light Meter HS1010A) whereas the other one was incubated side by side under the same light but tightly wrapped in aluminum foil. Plates were incubated simultaneously under these conditions for 4 days at 30°C and then imaged using an Epson flatbed scanner.

## Immunoblotting

Proteins were separated by SDS-PAGE and transferred to PVDF membranes, which were blocked with 5% (w:vol) non-fat dry milk and sequentially incubated with primary and secondary antibodies. For protein-protein binding experiments with GST-fused proteins, PVDF membranes were stained with Ponceau S and scanned before blocking. The primary antibodies used were the following: MYC (1:1,000), His (1:2,500), FLAG (1:2,000), α-tubulin (1:2,500), HA (1:1,000) and Gαs (1:500). The secondary antibodies were goat anti-rabbit Alexa Fluor 680 (1:10,000) and goat anti-mouse IRDye 800 (1:10,000). Infrared imaging of immunoblots was performed using an Odyssey Infrared Imaging System (Li-Cor Biosciences). Images were processed using the ImageJ software (NIH) and assembled for presentation using Photoshop and Illustrator softwares (Adobe).

## Acknowledgements

This work was supported by NIH grant R01GM136132 (to MG-M). MM was supported by an American Cancer Society - Funding Hope Postdoctoral Fellowship, PF-19-084-01-CDD. We thank N Lambert (Augusta University), K Martemyanov (The Scripps Research Institute, Florida), C Dessauer (University of Texas, Houston), J Sondek (University of North Carolina-Chapel Hill), P Wedegaertner (Thomas Jefferson University) , and T Meigs (University of North Carolina-Asheville) for providing the plasmids.

## Additional information

### Funding

| Funder | Grant reference number | Author |
| --- | --- | --- |
| National Institute of General Medical Sciences | R01GM136132 | Mikel Garcia-Marcos |
| American Cancer Society | PF-19-084-01-CDD | Marcin Maziarz |

The funders had no role in study design, data collection and interpretation, or the decision to submit the work for publication.

### Author contributions

Mikel Garcia-Marcos, Conceptualization, Formal analysis, Supervision, Funding acquisition, Investigation, Visualization, Methodology, Writing - original draft, Project administration, Writing - review and editing; Kshitij Parag-Sharma, Arthur Marivin, Marcin Maziarz, Formal analysis, Investigation, Writing - review and editing; Alex Luebbers, Formal analysis, Investigation; Lien T Nguyen, Investigation

### Author ORCIDs

Mikel Garcia-Marcos (iD) https://orcid.org/0000-0001-9513-4826
Kshitij Parag-Sharma (iD) https://orcid.org/0000-0003-3638-0941
Alex Luebbers (iD) http://orcid.org/0000-0001-7733-4250

### Decision letter and Author response

Decision letter https://doi.org/10.7554/eLife.60155.sa1
Author response https://doi.org/10.7554/eLife.60155.sa2

## Additional files

### Supplementary files

- Supplementary file 1. Sequences of LOV2GIV constructs.
- Transparent reporting form

### Data availability

All data generated or analysed during this study are included in the manuscript and supporting files. Source data files have been provided for Figures 2, 3 and 4 (and their corresponding supplements).

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

# Appendix 1

**Appendix 1—key resources table**

| Reagent type (species) or resource | Designation | Source or reference | Identifiers | Additional information |
|---|---|---|---|---|
| Strain, strain background (*Escherichia coli*) | BL21(DE3) | Invitrogen | Cat# C600003 | |
| Strain, strain background (*Escherichia coli*) | DH5alpha | New England Biolabs | Cat# C2987I | |
| Genetic reagent (*Saccharomyces cerevisiae*) | CY7967 [*MATα GPA1(1–41)-Gαi3 far1Δ fus1p-HIS3 can1 ste14:trp1:LYS2 ste3Δ lys2 ura3 leu2 trp1 his3*] | *Cismowski et al., 1999* | | Provided by James Broach (Penn State University) |
| Cell line (*Homo sapiens*) | HEK293T cells | ATCC | CRL3216 | |
| Antibody | α-tubulin (mouse monoclonal) | Sigma | T6074 | Immunoblotting Dilution (1: 2,500) |
| Antibody | FLAG tag (mouse monoclonal) | Sigma | F1804 | Immunoblotting Dilution (1: 2,000) |
| Antibody | His-tag (mouse monoclonal) | Sigma | H1029 | Immunoblotting Dilution (1: 2,500) |
| Antibody | Hemagglutinin (HA) tag (clone 12CA5) (mouse monoclonal) | Roche | Cat# 11583816001 | Immunoblotting Dilution (1: 1,000) |
| Antibody | MYC-tag (9B11) (mouse monoclonal) | Cell Signaling | Cat# 2276 | Immunoblotting Dilution (1: 1,000) |
| Antibody | Gαs (C-18) (rabbit polyclonal) | Santa Cruz Biotechnology | Cat# sc-383 | Immunoblotting Dilution (1: 500) |
| Antibody | Goat anti-rabbit Alexa Fluor 680 (goat polyclonal) | Life Technologies | Cat# A21077 | Immunoblotting Dilution (1:10,000) |
| Antibody | Goat anti-mouse IRDye 800 (goat polyclonal) | LiCor | Cat# 926–32210 | Immunoblotting Dilution (1:10,000) |
| Recombinant DNA reagent | pLIC-GST (plasmid) | *Cabrita et al., 2006* | | Provided by John Sondek (University of North Carolina- Chapel Hill) |
| Recombinant DNA reagent | pLIC-GST-LOV2GIV (plasmid) | This paper | | |
| Recombinant DNA reagent | pLIC-GST-LOV2GIV (L) (plasmid) | This paper | | Contains the dark-mimicking mutation C450S |
| Recombinant DNA reagent | pLIC-GST-LOV2GIV (D) (plasmid) | This paper | | Contains the dark-mimicking mutation C450S |

*Continued on next page*

*Appendix 1—key resources table continued*

| Reagent type (species) or resource | Designation | Source or reference | Identifiers | Additional information |
|---|---|---|---|---|
| Recombinant DNA reagent | pLIC-GST-LOV2GIVa (D) (plasmid) | This paper | | Contains the dark-mimicking mutation C450S |
| Recombinant DNA reagent | pLIC-GST-LOV2GIVb (D) (plasmid) | This paper | | Contains the dark-mimicking mutation C450S |
| Recombinant DNA reagent | pLIC-GST-LOV2GIVc (D) (plasmid) | This paper | | Contains the dark-mimicking mutation C450S |
| Recombinant DNA reagent | pLIC-GST-LOV2GIVd (D) (plasmid) | This paper | | Contains the dark-mimicking mutation C450S |
| Recombinant DNA reagent | pLIC-GST-LOV2GIVe (D) (plasmid) | This paper | | Contains the dark-mimicking mutation C450S |
| Recombinant DNA reagent | pLIC-GST-LOV2GIVe (L) (plasmid) | This paper | | Contains the lit-mimicking mutation I539E |
| Recombinant DNA reagent | pLIC-GST-LOV2GIVe (D) (plasmid) | This paper | | Contains the dark-mimicking mutation C450S |
| Recombinant DNA reagent | pLIC-GST-LOV2GIVe (L + FA) (plasmid) | This paper | | Contains the lit-mimicking mutation I539E and the GEF-deficient mutation F1685A |
| Recombinant DNA reagent | pET28b-Gαi3 (plasmid) | *Garcia-Marcos et al., 2009* | | For the bacterial expression of rat His-Gαi3 |
| Recombinant DNA reagent | pET28b-Gαo (plasmid) | *Garcia-Marcos et al., 2010* | | For the bacterial expression of rat His-Gαo (isoform A) |
| Recombinant DNA reagent | pLIC-His (plasmid) | *Stols et al., 2002* | | |
| Recombinant DNA reagent | pLIC-His- Gαi1 (int.6xHis) (plasmid) | This paper | | For the bacterial expression of rat Gαi1 with an internal His-tag |
| Recombinant DNA reagent | pET28b-Gαi2 (plasmid) | This paper | | For the bacterial expression of rat His-Gαi2 |
| Recombinant DNA reagent | p3XFLAG-CMV10-Gαi3 (plasmid) | *Garcia-Marcos et al., 2009* | | For the mammalian expression of rat FLAG-Gαi3 |
| Recombinant DNA reagent | pcDNA3.1(+)-Gαs (plasmid) | *Beas et al., 2012* | | For the mammalian expression of human Gαs |
| Recombinant DNA reagent | pcDNA3-Gαq-HA (plasmid) | *Wedegaertner et al., 1993* | | For the mammalian expression of mouse Gαq with an internal HA tag. Provided by P. Wedegaertner (Thomas Jefferson University) |

*Continued on next page*

*Appendix 1—key resources table continued*

| Reagent type (species) or resource | Designation | Source or reference | Identifiers | Additional information |
|---|---|---|---|---|
| Recombinant DNA reagent | pcDNA3.1-Gα12-MYC (plasmid) | *Ritchie et al., 2013* | | For the mammalian expression of mouse Gα12 with an internal MYC-tag. Provided by T. Meigs (University of North Carolina, Asheville) |
| Recombinant DNA reagent | pLIC-YES2 (plasmid) | *Coleman et al., 2016* | | |
| Recombinant DNA reagent | pLIC-YES2-N9Gpa1 (plasmid) | *Coleman et al., 2016* | | |
| Recombinant DNA reagent | pLIC-YES2-LOV2GIVe wt (plasmid) | This paper | | For the yeast expression of cytosolic LOV2GIVe |
| Recombinant DNA reagent | pLIC-YES2-LOV2GIVe (L) (plasmid) | This paper | | For the yeast expression of cytosolic LOV2GIVe bearing the lit-mimicking mutation I539E |
| Recombinant DNA reagent | pLIC-YES2-LOV2GIVe (D) (plasmid) | This paper | | For the yeast expression of cytosolic LOV2GIVe bearing the dark-mimicking mutation C450S |
| Recombinant DNA reagent | pLIC-YES2- N9Gpa1-LOV2GIVe wt (plasmid) | This paper | | For the yeast expression of membrane-anchored (m)LOV2GIVe |
| Recombinant DNA reagent | pLIC-YES2- N9Gpa1-LOV2GIVe (L) (plasmid) | This paper | | For the yeast expression of membrane-anchored (m)LOV2GIVe bearing the lit-mimicking mutation I539E |
| recombinant DNA reagent | pLIC-YES2- N9Gpa1-LOV2GIVe (D) (plasmid) | This paper | | For the yeast expression of membrane-anchored (m)LOV2GIVe bearing the dark-mimicking mutation C450S |
| Recombinant DNA reagent | pLIC-myc (plasmid) | *Cabrita et al., 2006* | | Provided by John Sondek (University of North Carolina- Chapel Hill) |
| Recombinant DNA reagent | pLIC-lyn11-myc (plasmid) | This paper | | |
| Recombinant DNA reagent | pLIC-lyn11-myc-LOV2GIVe (plasmid) | This paper | | For the mammalian expression of mLOV2GIVe |
| Recombinant DNA reagent | pLIC-lyn11-myc-LOV2GIVe (L) (plasmid) | This paper | | For the mammalian expression of mLOV2GIVe bearing the lit-mimicking mutation I539E |

*Continued on next page*

*Appendix 1—key resources table continued*

| Reagent type (species) or resource | Designation | Source or reference | Identifiers | Additional information |
|---|---|---|---|---|
| Recombinant DNA reagent | pLIC-lyn11-myc-LOV2GIVe (D) (plasmid) | This paper | | For the mammalian expression of mLOV2GIVe dark-mimicking mutation C450S |
| Recombinant DNA reagent | pLIC-lyn11-myc-LOV2GIVe (FA) (plasmid) | This paper | | For the mammalian expression of mLOV2GIVe GEF-deficient mutation F1685A |
| Recombinant DNA reagent | pLIC-lyn11-myc-LOV2GIVe (L) (FA) (plasmid) | This paper | | For the mammalian expression of mLOV2GIVe lit-mimicking mutation I539E and the GEF-deficient mutation F1685A |
| Recombinant DNA reagent | pcDNA3.1-Venus(155-239)-G$\beta_1$ (plasmid) | *Hollins et al., 2009* | | For the mammalian expression of G$\beta_1$ tagged with a fragment of Venus (VC-G$\beta_1$). Provided by N. Lambert (Augusta University, GA) |
| Recombinant DNA reagent | pcDNA3.1-Venus(1-155)-G$\gamma_2$ (plasmid) | *Hollins et al., 2009* | | For the mammalian expression of G$\gamma_2$ tagged with a fragment of Venus (VN-G$\gamma_2$). Provided by N. Lambert (Augusta University, GA) |
| Recombinant DNA reagent | pcDNA3.1-G$\beta_1$ (plasmid) | *Hollins et al., 2009* | | For the mammalian expression of untagged G$\beta_1$. Provided by N. Lambert (Augusta University, GA) |
| Recombinant DNA reagent | pcDNA3.1-G$\gamma_2$ (plasmid) | *Hollins et al., 2009* | | For the mammalian expression of untagged G$\gamma_2$. Provided by N. Lambert (Augusta University, GA) |
| Recombinant DNA reagent | pcDNA3-G$\alpha$i3 (plasmid) | *Garcia-Marcos et al., 2010* | | For the mammalian expression of rat G$\alpha$i3 |
| Recombinant DNA reagent | pcDNA3.1-masGRK3ct-Nluc (plasmid) | *Masuho et al., 2015* | | Provided by K. Martemyanov (Scripps Research Institute, FL) |
| Recombinant DNA reagent | pcDNA3.1-Nluc-EPAC-VV (plasmid) | *Masuho et al., 2015* | | Provided by K. Martemyanov (Scripps Research Institute, FL) |
| Recombinant DNA reagent | pcDNA3.1-3xHA-M4R (plasmid) | cDNA Resource Center at Bloomsburg University | Cat# MAR040TN00 | |

*Continued on next page*

*Appendix 1—key resources table continued*

| Reagent type (species) or resource | Designation | Source or reference | Identifiers | Additional information |
|---|---|---|---|---|
| Commercial assay or kit | QuikChange II Site-Directed Mutagenesis Kit | Agilent | Cat# #200523 | |
| Chemical compound, drug | NanoGlo Luciferase Assay System | Promega | Cat# N1120 | |
| Chemical compound, drug | Carbachol | Acros Organics | Cat# AC-10824 | |
| Chemical compound, drug | Atropine | Alfa Aesar | Cat# A10236 | |
| Chemical compound, drug | DL-isoproterenol hydrochloride | Alfa Aesar | Cat# J61788 | |
| Chemical compound, drug | $[\gamma\text{-}^{32}P]GTP$ | Perkin Elmer | NEG004Z250UC | |
| Chemical compound, drug | $[^{32}S]GTP\gamma S$ | Perkin Elmer | NEG030H250UC | |
| Chemical compound, drug | Fluorescein di-β-D-galactopyranoside (FDG) | Marker Gene Technologies | Cat# M0250 | |
| Software, algorithm | I-TASSER | *Yang et al., 2015* | https://zhanglab.ccmb.med.umich.edu/I-TASSER/ | Homology modeling server |
| Software, algorithm | ICM version 3.8–3 | Molsoft LLC., San Diego, CA | | Protein structure visualization software |

