## [Decision Letter]

**Acceptance summary:**

Methods to study the activity of heterotrimeric G-proteins are crucial to understand signalling via GPCRs. This new optogenetic tool that activates Gi proteins by blue light based on an engineered LOV2 domain provides a new method that will enhance our capabilities to investigate this important protein class.

**Decision letter after peer review:**

Thank you for submitting your article "Optogenetic activation of heterotrimeric Gi proteins by LOV2GIVe – a rationally engineered modular protein" for consideration by *eLife*. Your article has been reviewed by two peer reviewers, including Volker Dötsch as the Reviewing Editor and Reviewer #1, and the evaluation has been overseen by Jonathan Cooper as the Senior Editor.

The reviewers have discussed the reviews with one another and the Reviewing Editor has drafted this decision to help you prepare a revised submission.

Summary:

Garcia-Marcos et al. describe a method to study the activity of heterotrimeric G-proteins. These switches are usually activated via GPCRs and play very important roles in cellular signalling. Investigating their function is often difficult. Therefore, the authors have designed an optogenetic tool that activates Gi proteins by blue light based on an engineered LOV2 domain. They demonstrate that activation is specific and that the dark state has a much lower affinity than the light state. The optimization is quite impressive. Overall, this is an interesting and useful tool but some experimental verifications are required.

Essential revisions:

1) Figure 1 shows binding of the G protein to permanently on or off mutant versions of LOV2GIV. Since the G protein is purified, abundant and bound to GST-LOV2GIV, why is it not visible in the Ponceau S stained gel?

2) This figure needs additional controls. Is the interaction with WT LOV2GIV induced by light as shown in the cartoon? Does the interaction lead to increased GTP binding, as shown in the cartoon? Is the binding blocked by GIV residues known to be important for G protein binding as shown in the cartoon structure? Whether or not these controls have been used in the past, they should be done here as well for this particular fusion.

3) Figure 2A shows binding association (not dissociation as indicated) for the same constructs as in Figure 1. Figure 2B shows GTP hydrolysis but the function of GIV is to stimulate GTP binding, which is just as easy to measure. Again, this figure needs additional controls to show that it is activated by light and relies on key residues.

4) Figures 3 and 4 shows G protein activation in yeast and HEK293T cells. GIV leads to increased GTP binding but the cell assays do not measure G-α-GTP signaling but rather measure release of G-β-γ. A direct assay for G-α-GTP should be used. The yeast legend and figure do not match and the yeast assays in Figures 3 and 4 use different readouts when both could be used in parallel. A single concentration of a single agonist as a reference is not sufficient when the authors could easily do a concentration-response experiment with an antagonist as a negative control.

---

## [Author Response]

Summary:Garcia-Marcos et al. describe a method to study the activity of heterotrimeric G-proteins. These switches are usually activated via GPCRs and play very important roles in cellular signalling. Investigating their function is often difficult. Therefore, the authors have designed an optogenetic tool that activates Gi proteins by blue light based on an engineered LOV2 domain. They demonstrate that activation is specific and that the dark state has a much lower affinity than the light state. The optimization is quite impressive.Overall, this is an interesting and useful tool but some experimental verifications are required.

We appreciate the positive tone of the Summary of the review, which concludes that the new tool described here is interesting and useful. We have taken seriously the request for additional experimental verifications and tried our best to address it. For this, we have performed several new experiments that are presented in total of 7 panels distributed across 4 figures in the manuscript and one additional figure in this response.

Essential revisions:1) Figure 1 shows binding of the G protein to permanently on or off mutant versions of LOV2GIV. Since the G protein is purified, abundant and bound to GST-LOV2GIV, why is it not visible in the Ponceau S stained gel?

The reason why the G protein is not visible in the Ponceau S stained membrane is because it is masked by GST-LOV2GIV, which has almost the exact same apparent size as the G protein. We have now done experiments with a Gαi1 protein in which the N-terminus, which is not required for binding to GIV (de Opakua et al., 2017), is deleted and the N-terminal Histag has been cleaved after purification. Using this smaller construct, we can now see its binding in the Ponceau S stained membrane (Author response image 1).

**Author response image 1. sa2fig1:** Binding of N-terminally truncated Gαi1 (Gαi1ΔN) to LOV2GIVe is detectable by Ponceau S staining. Approximately 20 µg of the indicated purified GST or GST-LOV2GVe (L) were immobilized on glutathione-agarose beads and incubated with 3 µg (~300 nM) of purified Gαi3ΔN. Resin-bound proteins were eluted, separated by SDSPAGE, and analyzed by Ponceau S-staining and immunoblotting with the Gαi1/2 antibodies. One representative result of two independent experiments is shown.

Although we hope that this result will alleviate any concern of the reviewers, we have opted for not including it in the manuscript at this time because it might not be informative. However, we could do so if the reviewers disagree with our decision.

2) This figure needs additional controls. Is the interaction with WT LOV2GIV induced by light as shown in the cartoon? Does the interaction lead to increased GTP binding, as shown in the cartoon? Is the binding blocked by GIV residues known to be important for G protein binding as shown in the cartoon structure? Whether or not these controls have been used in the past, they should be done here as well for this particular fusion.

This point has several parts and we have carried out additional controls to address as many of them as possible:

– In the new Figure 1—figure supplement 1 we show that mutation of a residue of GIV known to be important for G protein binding (F1685) disrupts binding of Gαi3 to LOV2GIVe in the lit conformation.

– To assess GTP binding directly, we have now carried out GTPγS binding experiments. We found that LOV2GIVe in the lit conformation enhances GTPγS binding to Gαi3, and that this effect is inhibited by mutation of a residue of GIV known to be important for G protein binding (F1685). These results are shown in the new Figure 2—figure supplement 1.

The reviewers also pointed to the cartoon in Figure 1 that illustrates a model in which binding of LOV2GIVe to G proteins (and subsequent effects) can be triggered by light and asked for experiments to show directly that light induces LOV2GIVe binding to Gαi3. We would like to start by clarifying that our overall intention was to implement widely used and validated mutants of LOV2 that mimic either the lit or dark conformations to first characterize and optimize the biochemical properties of our construct, and then test it in functional cell-based assays with light. There are technical reasons for this overall approach that arise from our own expertise limitations. While we are comfortable with biochemistry and cell biology, our engineering abilities to create optogenetic hardware are quite limited. For this reason, we conducted experiments with light only for assays that were compatible with the technical resources available to us (repurposing of the fluorescence mode of a plate reader to provide a flash of light, or a homemade flat array of commercial blue LEDs for yeast assays).

We have not attempted to carry out the requested biochemical experiments with light because this would have required the creation and validation of new optogenetic hardware that we would have not been able to make/achieve in a reasonable amount of time. Under the current pandemic situation, reaching out to external collaborators is also not feasible. In other words, even under normal circumstances this would have taken us beyond the standard few months intended for revisions at *eLife*.

Although we acknowledge that not all the properties of LOV2GIVe have been tested with light, we believe that, taken collectively, the evidence provided in the manuscript strongly supports the conclusion that light-induced conformational changes in LOV2GIVe trigger G protein regulation. This has now been bolstered by the inclusion of controls with LOV2GIVe mutants that specifically disable G protein binding/regulation (see above and in the response to following review points below).

We have clarified in the text our overall approach of starting with biochemical characterization using mutants followed by validation in functional assays in cells with light (see subsection “Design and optimization of a photoswitchable G-protein activator”).

3) Figure 2A shows binding association (not dissociation as indicated) for the same constructs as in Figure 1.

We agree that Figure 2A shows association between LOV2GIVe and Gαi3. We believe that what might have led to confusion is the nomenclature “Kd” when we were actually reporting an *equilibrium* dissociation constant, which is frequently denoted as K_D_. Also, we believe that reporting equilibrium dissociation constants instead of equilibrium association constants is more widely used and more intuitive. To address this possible confusion, we have now changed the nomenclature from Kd to K_D_ in the text and in the figure.

Figure 2B shows GTP hydrolysis but the function of GIV is to stimulate GTP binding, which is just as easy to measure. Again, this figure needs additional controls to show that it is activated by light and relies on key residues.

As indicated in the response to point 2 above, we have carried out GTPγS binding experiments that show that LOV2GIVe in the lit conformation enhances GTP binding to Gαi3, and that this effect in inhibited by mutation of a residue of GIV known to be important for G protein binding (F1685) (see new Figure 2—figure supplement 1). For the same reasons explained in point 2, we have not carried out these biochemical experiments with light.

We would like to indicate that even though the original Figure 2B measured GTP hydrolysis, these measurements were done under steady-state conditions. Under these conditions, GTP hydrolysis reflects the rate of GTP binding because hydrolysis happens much faster (orders of magnitude) than nucleotide exchange (GTP binding) on Gαi. We have previously validated for GIV and other related GEFs that steady-state GTPase and GTPγS assays provide equivalent results. However, our first choice tends to be steady-state GTPase activity because of higher reproducibility (activity is linear for longer times) and because of the increasingly prohibitive cost of radio-labeled GTPγS. We have now modified the text to explain why steady-state GTPase assays with Gαi report the rate of nucleotide exchange (see subsection “LOV2GIVe binds and activates G-proteins efficiently in vitro only in its lit conformation”).

4) Figures 3 and 4 shows G protein activation in yeast and HEK293T cells. GIV leads to increased GTP binding but the cell assays do not measure G-α-GTP signaling but rather measure release of G-β-γ. A direct assay for G-α-GTP should be used.

We have now performed additional experiments to investigate the regulation of adenylyl cyclase, a classic effector of Gαi-GTP. We found that LOV2GIVe in the lit conformation leads to an *increase* of adenylyl cyclase activity as determined by the production of cAMP in response to isoproterenol stimulation (see new Figure 3—figure supplement 3). Although this is the opposite of what one would expect based on the effects of acute Gi activation, it is consistent with the expected effects of chronic Gi activation. It has been known for over 30 years that sustained activation of Gi leads to the phenomenon of adenylyl cyclase sensitization (Sharma et al., PNAS 1975 PMID: 1059094) – i.e., that agonists that activate adenylyl cyclase do so more robustly after chronic Gi activation (reviewed in Watts and Neve, 2005). Because expression of LOV2GIVe after transfection should exert its action for prolonged times, the observed increase in cAMP response is consistent with chronic Gi activation effects. This is further supported by inhibition of this potentiation effect upon introduction of a mutation in LOV2GIVe that prevents its ability to bind and activate Gαi3 (see new Figure 3—figure supplement 3).

The yeast legend and figure do not match and the yeast assays in Figures 3 and 4 use different readouts when both could be used in parallel.

Although we agree that the yeast assays in Figures 3 and 4 could theoretically be performed in parallel, there are technical reasons for our choice. The reason is suitability of each assay for the specific experimental setting. Figure 3 uses a β-galactosidase assay and Figure 4 uses a spot growth assay to measure transcriptional activity from the FUS1 promoter. The β-galactosidase assay is well suited to carry out quantitative measurements from cell cultures in suspension, but it is not well suited to achieve constant and uniform blue light illumination of cells. On the other hand, the spot growth assay is ideally suited to achieve constant and uniform illumination of cells growing on a flat surface. We have now modified the text to clarify the choice of yeast assays in the two figures (see subsection “LOV2GIVe activates G-protein signaling in cells upon illumination”).

We have also fixed the title of the Figure 3 legend.

A single concentration of a single agonist as a reference is not sufficient when the authors could easily do a concentration-response experiment with an antagonist as a negative control.

We are now showing an agonist concentration-response experiment, including also an antagonist, as a control for this assay (see new Figure 3—figure supplement 1). This result justifies not only the specificity of the readout based on complete blockade by the antagonist, but also the single concentration of agonist (100 µM) that was used in the comparison with LOV2GIVe in Figure 3B because it corresponds to the maximal response observed in the concentration-response curve.